# Theoretical and Experimental Investigation on the 3D Surface Roughness of Material Extrusion Additive Manufacturing Products

**DOI:** 10.3390/polym14020293

**Published:** 2022-01-11

**Authors:** Shijie Jiang, Ke Hu, Yang Zhan, Chunyu Zhao, Xiaopeng Li

**Affiliations:** 1School of Mechanical Engineering and Automation, Northeastern University, Shenyang 110819, China; hukehuk@163.com (K.H.); chyzhao@mail.neu.edu.cn (C.Z.); xpli@me.neu.edu.cn (X.L.); 2Key Laboratory of Dynamics and Reliability of Mechanical Equipment of Liaoning Province, School of Mechanical Engineering and Automation, Northeastern University, Shenyang 110819, China; 3Department of Cultural Foundation, Guidaojiaotong Polytechnic Institute, Shenyang 110023, China; 19218793@163.com

**Keywords:** material extrusion, surface roughness, bonding neck, theoretical model, experimental tests, sensitivity analysis

## Abstract

Material extrusion (ME), one of the most widely used additive manufacturing technique, has the advantages of freedom of design, wide range of raw materials, strong ability to manufacture complex products, etc. However, ME products have obvious surface defects due to the layer-by-layer manufacturing characteristics. To reveal the generation mechanism, the three-dimensional surface roughness (3DSR) of ME products was investigated theoretically and experimentally. Based on the forming process of bonding neck, the 3DSR theoretical model in two different directions (vertical and parallel to the fiber direction) was established respectively. The preparation of ME samples was then completed and a series of experimental tests were performed to determine their surface roughness with the laser microscope. Through the comparison between theoretical and experimental results, the proposed model was validated. In addition, sensitivity analysis is implemented onto the proposed model, investigating how layer thickness, extrusion temperature, and extrusion width influence the samples’ surface roughness. This study provides theoretical basis and technical insight into improving the surface quality of ME products.

## 1. Introduction

Additive manufacturing has gradually developed from the initial prototype manufacturing to direct manufacturing and mass manufacturing, which has a wide range of important application prospects [1,2,3]. Material extrusion (ME) is one of the most popular additive manufacturing technologies, which create three-dimensional (3D) solids through layer-by-layer manufacturing process [4,5]. It is inherently less wasteful than traditional subtractive methods of production and holds the potential to decouple social and economic value creation from the environmental impact of business activities [6]. However, due to the stratified nature of manufacturing characteristic, there is a big gap between ME products and those fabricated by traditional processing methods in terms of surface quality [7]. Since surface roughness affects the wear resistance, fatigue strength, and corrosion resistance of ME products, it needs to be as small as possible to maintain stability and extend the service life of the products for application in the field of automotive, electronics, aerospace, etc., [8].

Up to now, the surface roughness of ME products has been mainly assessed by performing iterative experiments at varying processing conditions. A lot of investigations on the optimization of processing parameters [9,10,11,12] have been performed to improve the surface quality of ME products, but the improvement is limited due to the small adjustment range. In order to effectively improve the surface quality of ME products and mitigate the time-consuming and expensive iterative experiments performed, it is necessary to clarify the generation mechanism of their surface roughness. Therefore, many scholars at home and abroad have carried out relevant research. Lalehpour et al. [13] proposed a theoretical model of surface roughness based on the material layer of ME products, and verified the model by experiments. The results showed that the proposed model could accurately predict the surface roughness of specific ME products. Vahabli et al. [14] proposed a new model on the basis of experimental research. Compared with other models [15,16], the proposed one had a significant improvement in the accuracy of surface roughness prediction. Angelo et al. [17], considering the influence of step effect on the surface profile, improved the model set up by Anh et al. [18,19], and proposed a new parameter Pa (ISO 4287) for evaluating the ME products’ surface quality. Comparing the predicted Pa value with the experimental data in the literature, it was found that the improved model was more accurate. Li et al. [20] proposed a data-driven model, which applied machine learning algorithms to accurately predict the surface roughness of ME products. Wang et al. [21] established a mechanism model based on the thermal analysis theory of bonding formation. The model could help to predict the surface roughness of printed heat-resistant parts. Kaji et al. [22] proposed an empirical model through actual observation and analysis of the geometry of the ME products’ surface profile. Li et al. [23] established a surface roughness model of ME products based on the surface profile representation method of parabolic curve and linear straight line. After comparing with experiments, they found that the model could improve the prediction accuracy. Assuming that the surface profile of ME products was parabolic, Pandey et al. [24] took into account the influence of layer thickness and printing direction, and created a semi-empirical formula for predicting the surface roughness of ME products. Combining theory and experience, Taufik et al. [25] established a theoretical model, which could simulate surface profile features of ME products.

Although the above scholars put forward different types of theoretical models or empirical formulas, they generally ignored the anisotropic characteristics of the ME product, i.e., the surface roughness varies in different directions. Furthermore, the surface roughness of only one side of the built product was taken into account, which was obviously limited for 3D entities. Besides, Vanaei et al. [26,27] studied the influence of extrusion temperature and the temperature between adjacent extruded filaments on the shape, size and fatigue life of ME products, separately. The experimental results showed that optimizing the above temperature was necessary for achieving the optimization of ME products’ forming quality. But little information is available in the literature mentioned above.

Due to the layer-by-layer manufacturing process, the forming quality of ME products was controlled by the thermal energy (temperature distribution) of the extruded material filaments [28,29]. The surface quality of ME products is related to the forming quality. The cross section of the material filament is elliptical, and bonding neck is formed between adjacent elliptical sections, which plays a vital role in the surface quality of ME products [30]. Therefore, based on the bonding neck forming process (horizontal and longitudinal bonding necks), a three-dimensional surface roughness (3DSR) theoretical model of ME products was established in this paper, with the anisotropic characteristics and temperature taken into account. Compared with a series of testing results, the proposed model was validated. In addition, sensitivity analysis of the model was carried out to predict how extrusion width, layer thickness, and extrusion temperature influence the ME products’ surface roughness.

## 2. Analytical Study

Due to the influence of gravity and the nozzle extrusion during the forming process, the cross section of the extruded filament of ME product is elliptical. There are two bonding necks generated in horizontal and longitudinal direction, respectively, as shown in Figure 1. It is assumed that the work done by the surface tension is equal to that by the viscous force when the bonding neck is formed between adjacent filaments, and the change in the length of the material filament is ignored. Based on the formation of bonding neck, there are four different types of surface roughness, which are the surface roughness vertical or parallel to the fiber direction based on the horizontal bonding neck (SRVF-HB and SRPF-HB) and the roughness vertical or parallel to the fiber direction based on the longitudinal bonding neck (SRVF-LB and SRPF-LB). Details are listed in Table 1.

### 2.1. Analytical Investigation Based on Horizontal Bonding Neck

#### 2.1.1. The Surface Roughness Vertical to the Fiber Direction

Figure 2 shows the schematic diagram of the formation stage of the horizontal bonding neck. In the forming process, the material flow is assumed to occur in a circle with the radius of the contact point being *r*_0_ (the radius of curvature of the ellipse).

At any time *t*, the instantaneous radius, bonding neck length, and angle are separately *r*_1_, 2*x,* and 2θ. They have the following relationship:(1)$$x=r_{1}\times \sin\theta$$

According to the principle of volume conservation, at any time *t*, the expression of the instantaneous radius *r*_1_ during the bonding process of adjacent material filaments is:(2)$$r_{1}=\frac{\sqrt{\pi }r_{0}}{\sqrt{\pi \;-\theta +\sin\theta \cos\theta }}$$
where $r_{0}$ is the radius of curvature of the ellipse, $r_{0}=a^{2}/b$, *a* is the semi-major axis of the ellipse, *b* is the semi-minor axis of the ellipse.

The net contact cross-sectional area is:(3)$$S\;=\frac{4\sqrt{\pi }lb^{2}}{a}\frac{\pi \;-\theta }{\sqrt{\pi \;-\theta +\sin\theta \cos\theta }}$$

Under the influence of surface tension [31],when material filaments are bonded, the work done by surface tension is:(4)$$W_{S}=-\Gamma \frac{dS}{dt},$$
where Γ is the surface tension coefficient, substituting Equation (3) into Equation (4), the work done by surface tension can be expressed as:(5)WS=4πlb2Γa(π−θ)cos2θ+sinθcosθ[(π −θ)+sinθcosθ]32θ·
where $\overset{\cdot }{\theta }$ is the rate of change of the instantaneous half-angle of the horizontal bonding neck, $\overset{\cdot }{\theta }=d\theta /dt$.

Assuming that the molten mass is the Newtonian fluid when adjacent extruded material filaments are bonded, the expression for the work done by viscous force of the Newtonian fluid [32] is:(6)$$W_{V}={∭}_{V}3\eta {\overset{\cdot }{\varepsilon }}^{2}dV$$
where η is the viscosity of the material, which is an important factor dependent on the extrusion temperature and the temperature between adjacent extruded filaments [26,27,33]. *V* is the bonded volume of adjacent extruded material filaments, $\overset{\cdot }{\varepsilon }$ is the fluid strain rate.

Assuming that the strain rate $\overset{\cdot }{\varepsilon }$ is continuous in the bonding area, it can be expressed as:(7)$$\overset{\cdot }{\varepsilon }=\frac{\partial v_{y}\left(A\right)}{\partial y}=\frac{v_{y}\left(A\right)-v_{y}\left(O\right)}{r_{1}}$$
where $v_{y}\left(O\right)$ is the flow rate of the material on the contact surface, $v_{y}\left(A\right)$ is the speed at which the center point of the material filaments bonding area moves to the contact point, $v_{y}\left(A\right)$ can be expressed as:(8)vy(A)=(θ−π) πrb2sinθa (π −θ+sinθcosθ)32θ·

Substituting Equation (8) into Equation (7), the expression of the strain rate is given by:(9)$$\overset{\cdot }{\varepsilon }=\frac{\left(\pi \;-\theta \right)\;\sin\theta }{\left(\pi \;-\theta +\sin\theta \cos\theta \right)}\overset{\cdot }{\theta }$$

Substituting Equation (9) into Equation (6), the expression of the work done by the viscous force is:(10)$$W_{V}=\frac{6\pi l\eta b^{4}}{a^{2}}\;\frac{{\left(\pi \;-\theta \right)}^{2\;}\sin^{2}\theta }{{\left(\pi \;-\theta +\sin\theta \cos\theta \right)}^{2}}{\overset{\cdot }{\;\theta }}^{2}$$

Let the work done by surface tension *W_S_* equals to work done by viscous force *W_V_*, the expression of the rate of change of the instantaneous half-angle to the horizontal bonding neck can be given by:(11)$$\overset{\cdot }{\theta }=\frac{2\Gamma a\;{\left(\pi \;-\theta +\sin\theta \cos\theta \right)}^{1/2}}{3\sqrt{\pi }b^{2}\eta \;{\left(\pi \;-\theta \right)}^{2\;}\sin^{2}\theta }\times \left[\left(\pi \;-\;\theta \right){\;\cos}^{2}\theta \;+\;\sin\theta \cos\theta \right]$$

Substituting Equation (2) into Equation (1), the expression of the half-length to the horizontal bonding neck is:(12)$$x=\frac{\sin\theta \sqrt{\pi }b^{2}}{a\sqrt{\pi \;-\theta +\sin\theta \cos\theta }}$$

Using the initial conditions $\theta \left(0\right)=\theta_{0\;}=0$ to solve Equation (11) to obtain the instantaneous half-angle of the horizontal bonding neck at a certain moment. Then substituting the half-angle θ into Equation (12) to obtain the horizontal bonding neck.

During the bonding process of adjacent filaments, the bonding neck stops growing when the melt temperature of the extrudate drops to the critical temperature. Therefore, it is necessary to analyze the extrudate’s cooling time.

Bellehumeur et al. [34] used the lumped capacity (LC) analysis for modeling the cooling process of the extrudate and thus proposed the cooling model as follows:(13)$$T=T_{E}+\left(T_{L}-T_{E}\right)\times e^{-mx},$$
with
(14)$$m=\frac{\sqrt{1+4\alpha \beta }-1}{2\alpha }\;\mathrm{and}\;x=vt$$
where
(15)$$\alpha \;=\frac{k}{\rho Cv}\;\mathrm{and}\;\beta =\frac{h_{1}P}{\rho CAv}$$
where $T_{E}$ is the envelope temperature of the environment, $T_{L}$ is the melting temperature, *T* is the real-time temperature, ρ is the density of the material filament, *k* is the thermal conductivity of the material, *h*_1_ is the system convection coefficient. The terms *C*, *A,* and *P* separately represent the specific heat capacity of the material, area, and perimeter of the elliptical section.

According to the actual situation of the formation of the horizontal bonding neck, the model schematic of the surface roughness vertical to the fiber direction based on horizontal bonding neck (SRVF-HB) is determined, as shown in Figure 3.

The (*xoy*) coordinate system is established based on the cross section of the *n*th layer of material filament and its center point, and the cross-sectional profile can be expressed as:(16)$$\frac{x^{2}}{a^{2}}+\frac{y^{2}}{b^{2}}=1$$

Extrusion width *E* and layer thickness *K* can be given respectively by:*E* = 2*a* and *K* = 2*b*.(17)

Substituting Equation (12) into Equation (16), the coordinates of $P_{1}\left(x_{1},y_{1}\right)$ and $P_{2}\left(x_{2},y_{2}\right)$ can be obtained, thus the width of the overlapping area *c* is:(18)$$c=2b\;\left(1-\sqrt{1-\frac{\pi b^{4}\sin^{2}\theta }{\pi \;-\theta +\sin\theta \cos\theta }}\right)$$

The boundary is determined by two lines parallel to the surface normal vector, which are expressed as:(19)$$y_{l_{b1}}=y_{1}$$
(20)$$y_{l_{b2}\;}=y_{2}$$

The temporary centerline $l_{t}$ is located between the straight line $l_{v}$ and $l_{p}$, and the three straight lines are parallel to each other. Let the intersection of the line $l_{t}$ and $l_{b1}$ be $P_{t1}\left(x_{t1},y_{t1}\right)$, and the expression of $x_{t1}$ is:(21)$$x_{t1\;}=x_{t1\_\mathrm{old}}+d\left(A_{P}>A_{V}\right)$$
where *d* is iterative increment. Hence the expression of $l_{t}$ is:(22)$$x_{l_{t}}=x_{t1}$$

Combining Equation (16) with Equation (22), and Equation (20) with Equation (22) respectively can obtain the intersections $P_{t2}\;\left(x_{t2},y_{t2}\right)$, $P_{t3}\;\left(x_{t3},y_{t3}\right),$ and $P_{t4}\;\left(x_{t4},y_{t4}\right)$.

The enclosed areas $A_{P}$ and $A_{V}$ are composed of straight lines $l_{t}$, peak profiles, valley profiles, and their boundaries. The enclosed area $A_{P}$ and $A_{V}$ can be expressed as:(23)$$A_{P\;}=\int_{y_{t3}}^{y_{t2}}\;\left[f\;\left(y\right)+l_{t}\;\left(y\right)\right]\;dy$$
(24)$$A_{V1}=\;\int_{x_{1}}^{x_{t1}}\left[l_{b1}\;\left(x\right)-f\;\left(x\right)\right]\;dx\;+\int_{x_{t1}}^{x_{t2}}\left[l_{t}\;\left(x\right)-f\;\left(x\right)\right]\;dx$$
(25)$$A_{V2}=\int_{x_{2}}^{x_{t3}}\left[f\;\left(x\right)-l_{b2\;}\left(x\right)\right]\;dx\;+\int_{x_{t3}}^{x_{t4}}\left[l_{t}\;\left(x\right)-l_{b2}\;\left(x\right)\right]\;dx$$
(26)$$A_{V}=A_{V1}+A_{V2}$$
where f(x), f(y), $l_{b1}\left(x\right)$, $l_{b2}\left(x\right)$, $l_{t}\left(x\right),$ and $l_{t}\left(y\right)$ are the expansion formulas of elliptic curve, boundary line, and temporary centerline with respect to *x* and *y* separately.

When $l_{t}$ is in a certain position so that $A_{P}=A_{V}$, the expression of the straight line $l_{t}$ can be obtained by substituting Equation (9) to Equation (12). At this time, $l_{t}$ is the arithmetic mean centerline of the contour.

Considering the influence of forming accuracy on the surface roughness of the ME sample, the height change of the surface profile vertical to the fiber direction can be expressed as:*h* = *qs*(27)
where *s* is forming accuracy, s=±0.1 mm, *q* is the influence coefficient.

Therefore, according to the definition of surface roughness [18], SRVF-HB can be expressed as:(28)$$R_{a,V-\mathrm{HB}}=\frac{\sum_{i=1}^{n}\left|l_{i}\;\pm \;qs\right|}{n}$$
where *n* is the number of sampling points, $n=\frac{a-x_{1}}{m}+1$, *m* is the interval distance between sampling points, *m* = 0.001 mm.

#### 2.1.2. The Surface Roughness Parallel to the Fiber Direction

Ideally, the surface profile parallel to the fiber direction of the ME products should have been smooth. As a matter of fact, it is an undulating curve, with the schematic diagram shown in Figure 4.

The maximum height of the surface profile depends on the height change of the surface profile vertical to the fiber direction, and the expression is:*r* = *h*(29)

Based on the SRVF-HB, assuming that *u* is the width of the surface profile parallel to the fiber direction, the surface roughness parallel to the fiber direction based on horizontal bonding neck (SRPF-HB) can be determined by:(30)$$R_{a,P-\mathrm{TB}}=\frac{1}{l}\int \left|f\left(x\right)\right|dx$$
*l* = *u*(31)
(32)$$\left|f\left(x\right)\right|=X_{1}+X_{2}+X_{3}$$

Substituting Equation (30) into Equation (32), the expression of $R_{a,P-\mathrm{HB}}$ is:(33)$$R_{a,P-\mathrm{HB}}=\frac{X_{1}+X_{2}+X_{3}}{u}$$
where *X*_1_, *X*_2_, and *X*_3_ are the areas of the enclosed section.

### 2.2. Analytical Investigation Based on Longitudinal Bonding Neck

#### 2.2.1. The Surface Roughness Vertical to the Fiber Direction

Figure 5 is a schematic diagram of the formation stage of the longitudinal bonding neck. The formation principle is similar to that of the horizontal bonding neck.

At any time *t*, the relationship between the instantaneous radius $r_{2}$, longitudinal bonding neck length 2*y,* and instantaneous angle 2β is:(34)$$y=r_{2}\times \sin\beta$$

According to the principle of volume conservation, the expression of the instantaneous radius *r*_2_ during the bonding process of adjacent material filaments is:(35)$$r_{2}=\frac{\sqrt{\pi }r_{0}}{\sqrt{\pi \;-\beta +\sin\beta \cos\beta }}$$

Similarly, the expression of the instantaneous half-angle change rate of the longitudinal bonding neck is:(36)$$\overset{\cdot }{\beta }\;=\;\frac{2\Gamma a\;{\left(\pi \;-\beta +\sin\beta \cos\beta \right)}^{1/2}}{3\sqrt{\pi }b^{2}\eta \;{\left(\pi -\beta \right)}^{2}\sin^{2}\beta }\times \left[\left(\pi \;-\beta \right){\;\cos}^{2}\beta +\sin\beta \cos\beta \right]$$

Substituting Equation (33) into Equation (32), the expression of the half-length to the longitudinal bonding neck is:(37)$$y=\frac{\sin\beta \sqrt{\pi }b^{2}}{a\sqrt{\pi \;-\beta +\sin\beta \cos\beta }}$$

Using the initial conditions $\beta \left(0\right)=\beta_{0\;}=0$ to solve Equation (36) to obtain the instantaneous half-angle of the longitudinal bonding neck at a certain moment. Then substituting the half-angle β into Equation (37) to determine the longitudinal bonding neck.

According to the actual situation of the longitudinal bonding neck, the model schematic of the surface roughness vertical to the fiber direction based on longitudinal bonding neck (SRVF-LB) is determined, as shown in Figure 6.

Similarly to Equation (28), SRVF-LB can be expressed as:(38)$$R_{a,V-\mathrm{LB}}=\frac{\sum_{i=1}^{n}\left|l_{i}\;\pm \;qs\right|}{n}$$

#### 2.2.2. The Surface Roughness Parallel to the Fiber Direction

As shown in Figure 7, based on the SRVF-LB, assuming that *v* is the width of the surface profile parallel to the fiber direction, the surface roughness parallel to the fiber direction based on longitudinal bonding neck (SRPF-LB) can be similarly given by:(39)$$R_{a,P-\mathrm{LB}}=\frac{{X^{\prime }}_{1}+{X^{\prime }}_{2}+{X^{\prime }}_{3}}{v}$$
where ${X^{\prime }}_{1}$, ${X^{\prime }}_{2}$, and ${X^{\prime }}_{3}$ are the areas of the enclosed section.

## 3. Experimental Analysis

### 3.1. Sample Preparation

To determine the surface roughness of ME products, cube samples with the size of 20 × 20 × 20 mm were fabricated by the ME device (FLSUN-QQ, Chaokuo, China), as shown in Figure 8. All the processing parameters were set the same, such as printing speed, layer thickness, extrusion width, build direction, extrusion temperature, etc. Details are shown in Table 2. The sample material is polylactic acid (PLA), which is biodegradable with the characteristics of good thermoplasticity, high strength, and excellent processing performance, etc., [35].

### 3.2. Surface Roughness Test

The experimental study of SRVF-HB/LB and SRPF-HB/LB of each sample was carried out by using the 3D laser microscope (LEXT OLS4100, Olympus, Tokyo, Japan), as shown in Figure 9. The standard for measuring surface roughness is ISO 4287: 1997. Through microscope observation, the surface roughness test was carried out on different typical positions (where the filaments bonded normally and no defects occurred) of the sample’s surface [36]. Taking into account the layer-by-layer forming process and orthotropic characteristics, each sample contains two sides with different surface roughness, and each side has two different surface roughness (separately SRVF-HB/LB and SRPF-HB/LB). For each sample, three different typical positions for each side in each direction were randomly selected for testing to ensure the representativeness and repeatability of the results. Each position was measured ten times, and there were totally 60 sets of surface roughness results obtained for each sample, separately including 15 sets for SRVF-HB, SRVF-LB, SRPF-HB, and SRPF-LB. Therefore, a total of 180 sets of tests were completed on the three samples. In order to ensure the accuracy and reliability of the test results, the average value of the data measured on each side in each direction was taken as the analysis result.

## 4. Results and Discussions

### 4.1. Results Based on the Horizontal Bonding Neck

#### 4.1.1. Surface Roughness Vertical to the Fiber Direction (SRVF)

Based on the horizontal bonding neck, the theoretical and experimental surface profiles of the ME sample ($R_{i}$, *I* = 1~3) vertical to the fiber direction are shown in Figure 10. From the comparison, it can be seen that the theoretical results are in good agreement with the experimental ones. Therefore, the SRVF-HB theoretical model can accurately predict the surface profile of the ME sample vertical to the fiber direction based on the horizontal bonding neck.

Table 3 summarizes the detailed results of ME samples ($R_{i}$, *i* = 1~3). It can be seen that the discrepancy between the theoretical and experimental results is in the range of 1.48~7.42%, validating the correctness of the theoretical model. Therefore, the proposed model can accurately predict the ME samples’ SRVF-HB.

#### 4.1.2. Surface Roughness Parallel to the Fiber Direction (SRPF)

Figure 11 shows the sample’s theoretical and average experimental results of the surface profile parallel to the fiber direction. It can be seen that the predictions generally agree well with the measurements. Therefore, the proposed SRPF-HB model can give reliable predictions on the surface profile parallel to the fiber direction based on the horizontal bonding neck.

The predicted and measured surface roughness of ME samples are detailed in Table 4. It can be seen that the discrepancy between theoretical and experimental results is small (4.13~7.82%). Therefore, the proposed model can accurately predict SRPF-HB of ME samples.

### 4.2. Results Based on the Longitudinal Bonding Neck

#### 4.2.1. Surface Roughness Vertical to the Fiber Direction (SRVF)

Figure 12 presents the theoretical and experimental results of the surface profile vertical to the fiber direction of the ME samples ($R_{i}$, *i* = 1~3). It can be seen that both the values and trend are in good agreement. Therefore, the SRVF-LB model can accurately predict the surface profile vertical to the fiber direction based on the longitudinal bonding neck.

Table 5 lists the average experimental and theoretical results of ME samples’ SRVF-LB. The discrepancy is only in the range of 1.86~4.50%. Therefore, the proposed model can accurately predict the results of SRVF-LB.

#### 4.2.2. Surface Roughness Parallel to the Fiber Direction (SRPF)

As shown in Figure 13, the predicted surface profile of the sample parallel to the fiber direction based on the longitudinal bonding neck is in generally good agreement with the average measured results. Therefore, the proposed SRPF-LB model can give a reliable prediction on the surface profile of the ME sample parallel to the fiber direction based on the longitudinal bonding neck.

Table 6 compares the theoretical and average experimental results of the samples’ SRPF-LB. It can be seen that they are in good agreement, and the proposed model is reliable.

## 5. Sensitivity Analysis

In order to analyze the sensitivity of the proposed model, three key processing parameters were selected for research, as shown in Table 7. Each parameter was individually modified to the value shown in Case 1~3, and the other processing parameters remain unchanged. By comparing the predicted surface roughness values under different processing parameters, the sensitivity of the model can be determined.

### 5.1. Effect of Extrusion Width

Figure 14 and Figure 15 show the influence of different extrusion widths (0.2, 0.3, and 0.4 mm) on the predicted values of SRVF-HB, SRPF-HB, SRVF-LB, and SRPF-LB of ME products. For SRVF-HB, when the extrusion width is increased from 0.2 to 0.4 mm, the corresponding predicted values are reduced by 13.17% (from 28.10 down to 24.40 μm); while for SRPF-HB, it is reduced by 7.45%, with the value decreasing from 9.67 to 8.95 μm; taking into account SRVF-LB, its value is increased by 8.45% (from 29.60 up to 32.20 μm) when the extrusion width is increased from 0.2 to 0.4 mm; about SRPF-LB, it is increased to 5.80 μm. This can be explained that increasing the extrusion width alone can increase the length of the horizontal bonding neck and slightly reduce the longitudinal one, which thereby reduces the height of the surface profile (or surface roughness) based on the horizontal bonding neck, but vice versa for that based on the longitudinal bonding neck.

### 5.2. Effect of Layer Thickness

Figure 16 and Figure 17 present the comparison of predicted values of SRVF-HB, SRPF-HB, SRVF-LB, and SRPF-LB of ME products built with different layer thickness, which is 0.1, 0.15, and 0.2 mm. For SRVF-HB, when the layer thickness is increased from 0.1 to 0.2 mm, the predicted values are increased by 56.74% (from 21.50 up to 33.70 μm). While for SRPF-HB, it is increased by 23.72%, with the value increased to 10.17 μm. Taking into account SRVF-LB, its value is reduced by 29.82% (from 38.90 down to 27.30 μm) when the layer thickness is increased from 0.1 to 0.2 mm; about SRPF-LB, it is dropped to 5.60 μm. Similarly, this is because as the layer thickness increases, the horizontal bonding neck becomes slightly smaller, resulting in an increase in the surface roughness based on the horizontal bonding neck, while vice versa for the surface roughness based on the longitudinal bonding neck.

### 5.3. Effect of Extrusion Temperature

The effect of different extrusion temperature (190, 200, 210 °C) on ME products’ surface roughness is shown in Figure 18 and Figure 19. In general, the surface roughness of ME samples will be decreased when the extrusion temperature increases. For SRVF-HB, when the extrusion temperature is increased from 190 to 210 °C, the corresponding predicted values are reduced by 35.15% (from 29.30 down to 19.00 μm); while for SRPF-HB, it is reduced by 29.21%, with the value being 7.41 μm. Taking into account SRVF-LB, its value is decreased by 27.81% (from 35.60 down to 25.70 μm) when the extrusion temperature is increased from 190 to 210 °C; about SRPF-LB, it is reduced to 5.20 μm. This can be explained that increasing the extrusion temperature will reduce the melt viscosity and increase the cooling time of the extrudate, and thereby increase the bonding neck, which lowers the height of surface profile, and thus betters the surface quality.

### 5.4. Influencing Degree of the Processing Parameters

Taking SRVF-HB as an example, when the extrusion width is doubled, the predicted surface roughness will be decreased by 13.17%. For the layer thickness, the surface roughness will be increased by 56.74%. Besides, when the extrusion temperature is increased by 20 °C, the predicted surface roughness will be reduced by 35.15%. Details are shown in Table 8. Therefore, among the three processing parameters, the layer thickness and extrusion temperature have more significant effect on the surface roughness of ME products, and least for the ex-trusion width.

## 6. Conclusions

Based on the bonding neck forming process, a theoretical model of the three-dimensional surface roughness (3DSR) of ME products is established considering its actual surface conditions. Through the comparison between predictions and measurements, the proposed model is validated and it can give reliable predictions in the ME samples’ 3DSR (i.e., SRVF-HB/LB and SRPF-HB/LB), with the discrepancy under 10.52% between predicted and measured results. In addition, the sensitivity analysis shows that increasing the extrusion width will decrease SRVF-HB/SRPF-HB, and increase SRVF-LB/SRPF-LB; but the layer thickness is in vice versa. When the extrusion temperature is decreased, the SRVF-HB/LB and SRPF-HB/LB will all be reduced. Among the three processing parameters, the effect of layer thickness and extrusion temperature is more significant on the 3DSR of ME products. The proposed model can help mitigate the time-consuming and expensive iterative experiments performed, and it can also provide the clue to find the effective method to improve the surface quality of ME components.

## Figures and Tables

**Figure 1 polymers-14-00293-f001:**
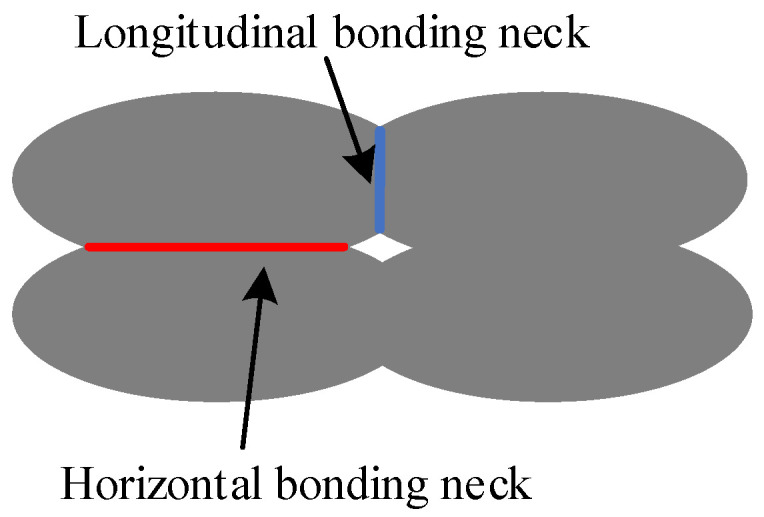
Schematic diagram of bonding neck.

**Figure 2 polymers-14-00293-f002:**
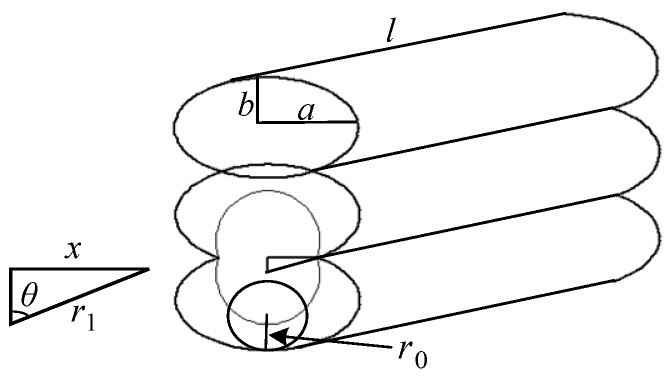
Schematic diagram of the formation of the horizontal bonding neck.

**Figure 3 polymers-14-00293-f003:**
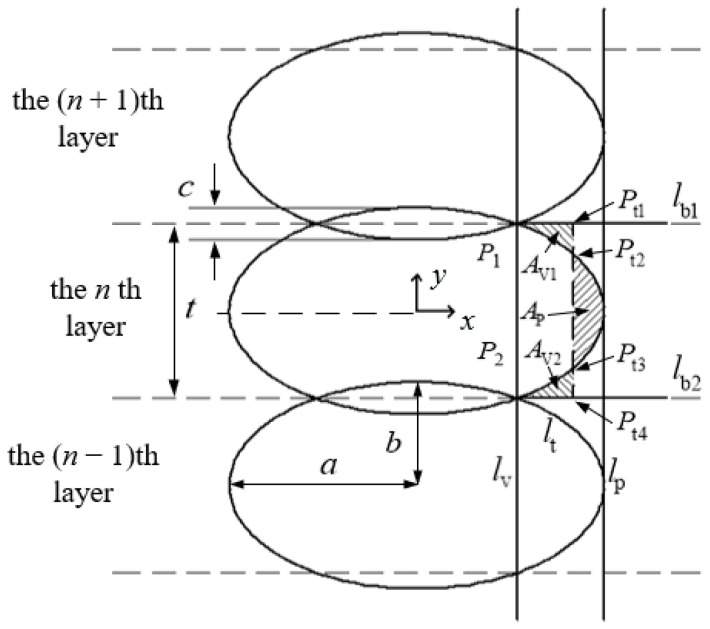
SRVF-HB model schematic.

**Figure 4 polymers-14-00293-f004:**
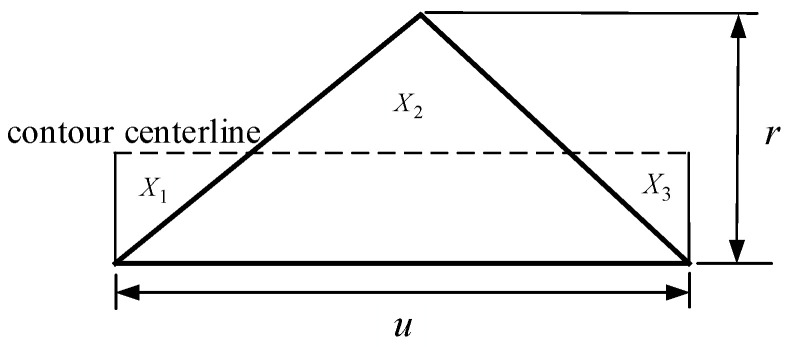
SRPF-HB model schematic.

**Figure 5 polymers-14-00293-f005:**
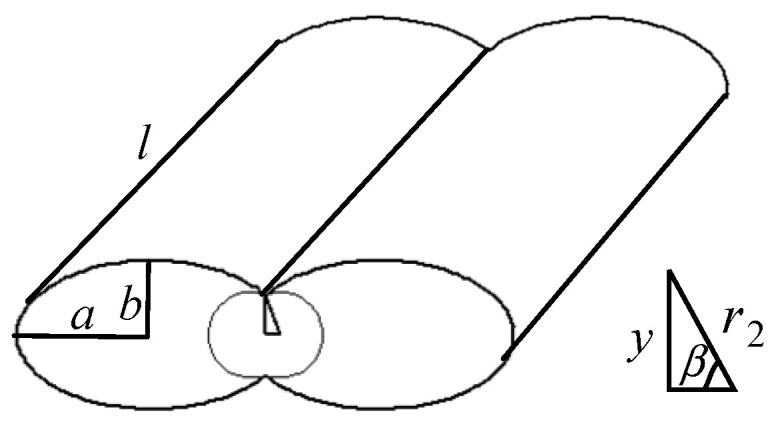
Schematic diagram of the formation of the longitudinal bonding neck.

**Figure 6 polymers-14-00293-f006:**
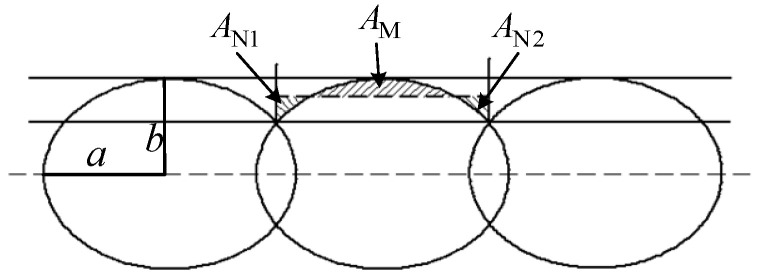
SRVF-LB model schematic.

**Figure 7 polymers-14-00293-f007:**
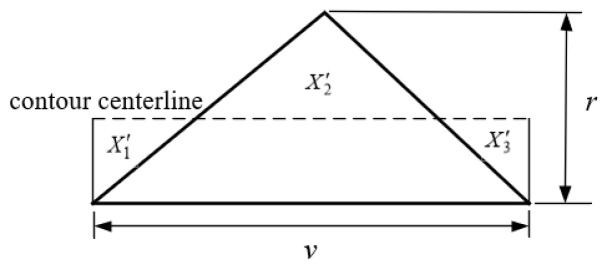
SRPF-LB model schematic.

**Figure 8 polymers-14-00293-f008:**
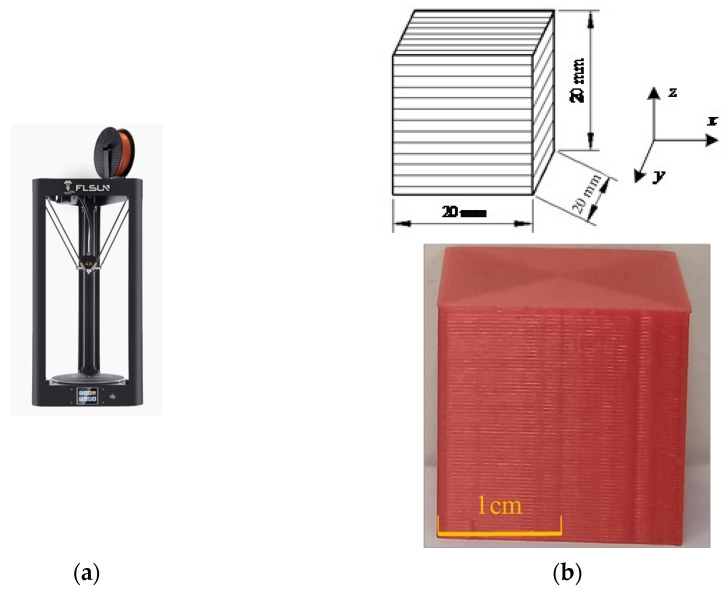
The ME equipment and testing samples: (**a**) ME device; (**b**) three-dimensional drawing of the samples.

**Figure 9 polymers-14-00293-f009:**
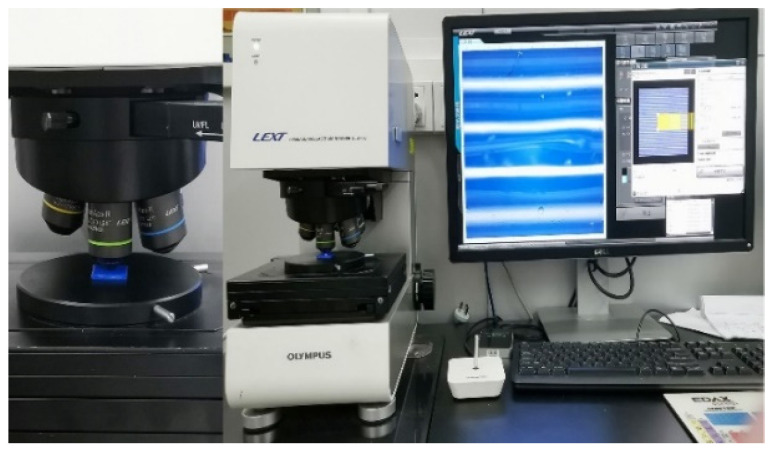
The surface roughness test equipment and samples.

**Figure 10 polymers-14-00293-f010:**
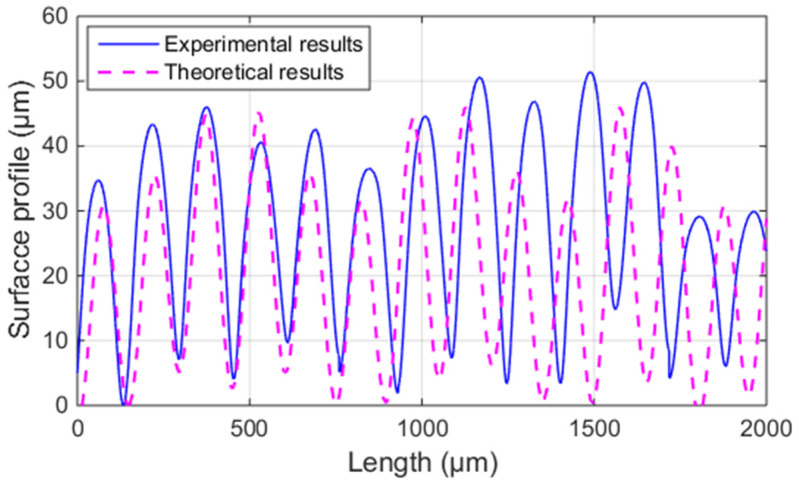
Theoretical and average experimental results of the surface profile of the ME samples vertical to the fiber direction.

**Figure 11 polymers-14-00293-f011:**
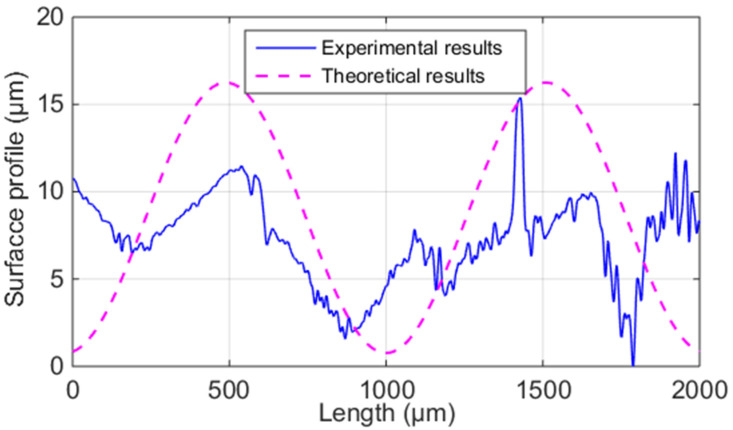
Theoretical and average experimental results of the surface profile of the ME samples parallel to the fiber direction.

**Figure 12 polymers-14-00293-f012:**
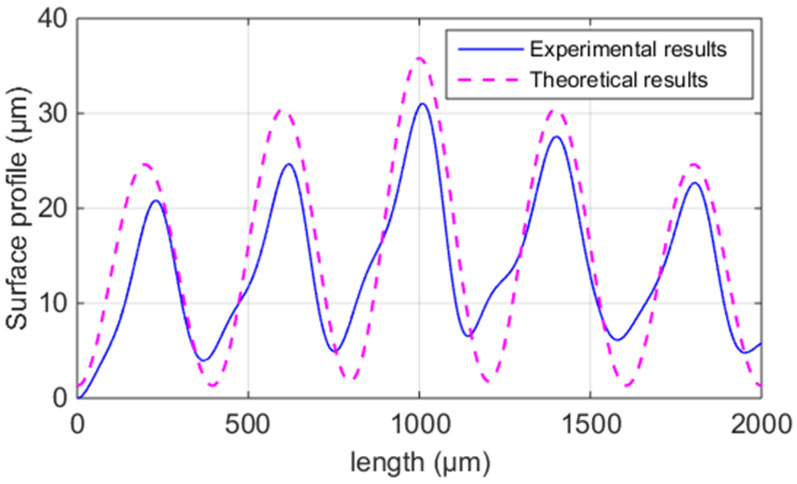
Theoretical and average experimental results of the surface profile of the ME samples vertical to the fiber direction.

**Figure 13 polymers-14-00293-f013:**
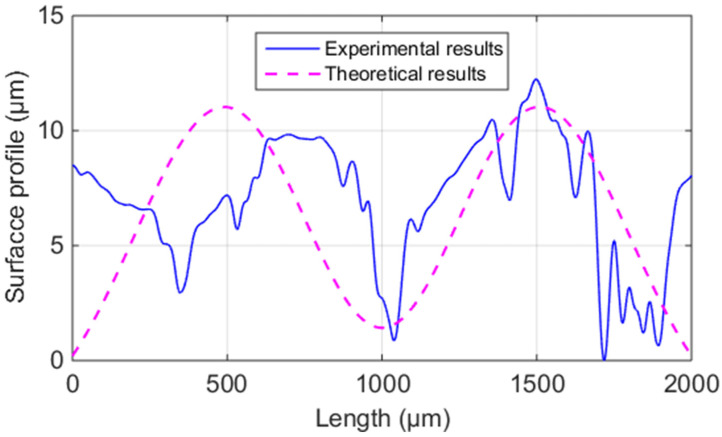
Theoretical and average experimental results of the surface profile of the ME samples parallel to the fiber direction.

**Figure 14 polymers-14-00293-f014:**
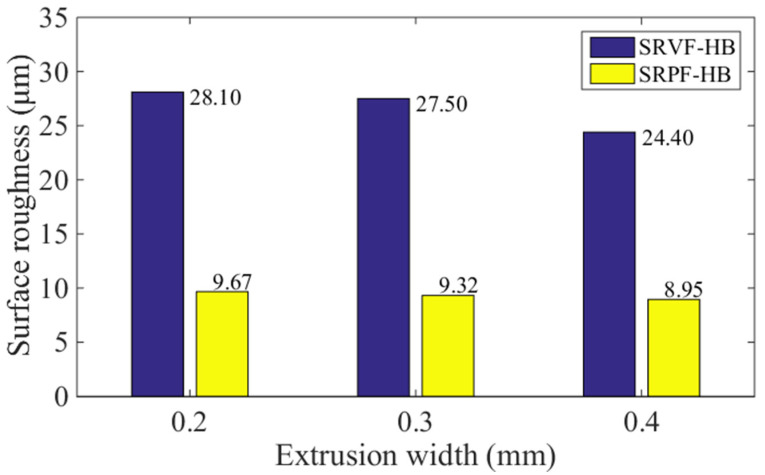
Effect of different extrusion width on the SRVF-HB and SRPF-HB.

**Figure 15 polymers-14-00293-f015:**
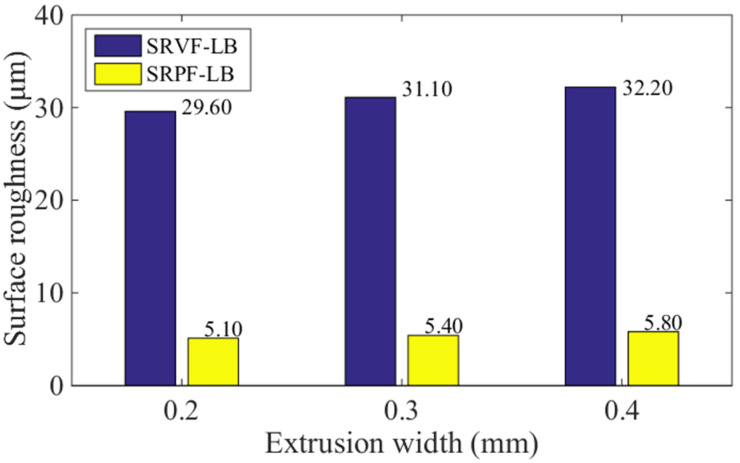
Effect of different extrusion width on the SRVF-LB and SRPF-LB.

**Figure 16 polymers-14-00293-f016:**
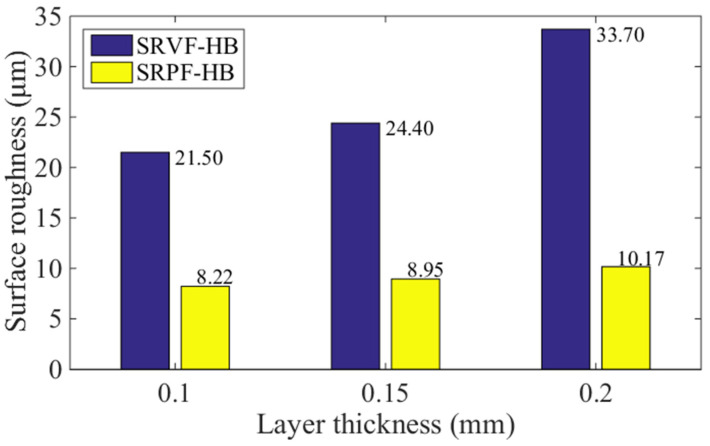
Effect of different layer thickness on the SRVF-HB and SRPF-HB.

**Figure 17 polymers-14-00293-f017:**
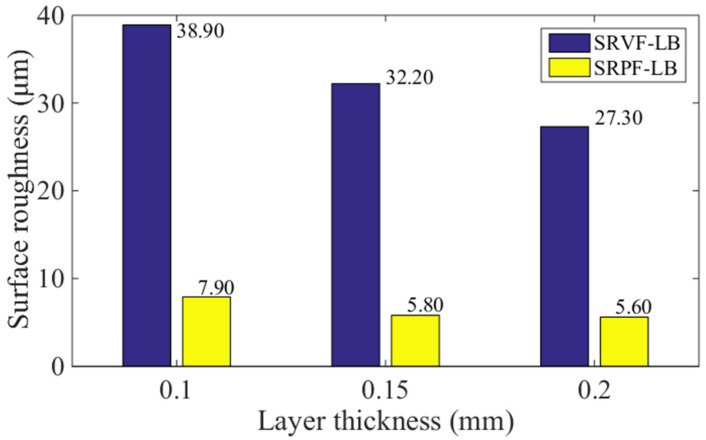
Effect of different layer thickness on the SRVF-LB and SRPF-LB.

**Figure 18 polymers-14-00293-f018:**
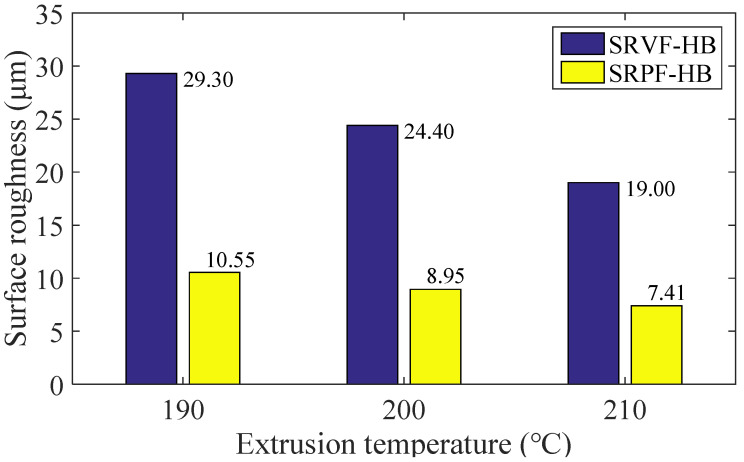
Effect of different extrusion temperature on the SRVF-HB and SRPF-HB.

**Figure 19 polymers-14-00293-f019:**
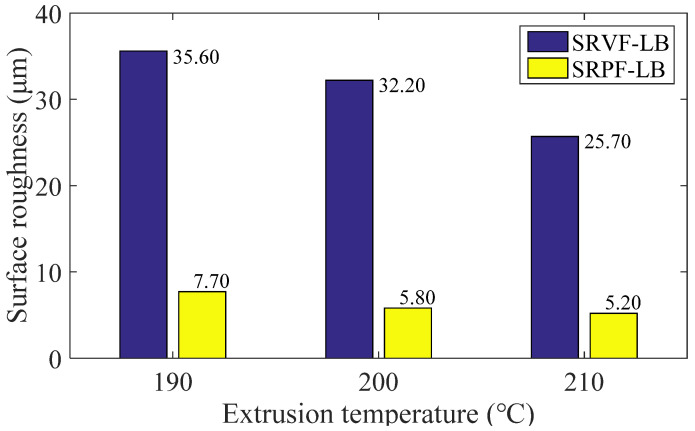
Effect of different extrusion temperature on the SRVF-LB and SRPF-LB.

**Table 1 polymers-14-00293-t001:** Four different types of surface roughness.

Category	Schematic Diagram
Based on horizontal bonding neck	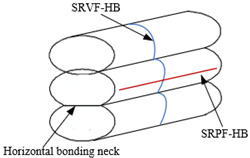
Based on longitudinal bonding neck	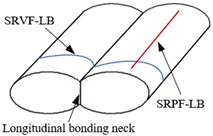

**Table 2 polymers-14-00293-t002:** Processing parameter settings.

Sample (*I* = 1~3)	$R_{i}$
Extrusion width (mm)	0.4
Layer thickness (mm)	0.15
Extrusion temperature (°C)	200
Build direction	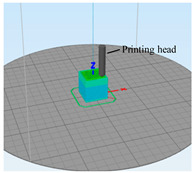
Schematic cross section	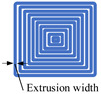
Printing speed (mm $s^{-1}$)	60
Platform Temperature (°C)	60

**Table 3 polymers-14-00293-t003:** The detailed results of SRVF-HB.

Sample (*z* = 1~10)	Average Measurements (μm)	Prediction (μm)	Errors (%)
$R_{1}\_z$	22.59	24.4	7.42
$R_{2}\_z$	23.28		4.59
$R_{3}\_z$	24.07		1.48

**Table 4 polymers-14-00293-t004:** The detailed results of SRPF-HB.

Sample (*z* = 1~10)	Average Measurements (μm)	Prediction (μm)	Errors (%)
$R_{1}\_z$	8.25	8.95	7.82
$R_{2}\_z$	8.58		4.13
$R_{3}\_z$	9.34		4.36

**Table 5 polymers-14-00293-t005:** The detailed results of SRVF-LB.

Sample (*z* = 1~10)	Average Measurements (μm)	Prediction (μm)	Errors (%)
$R_{1}\_z$	31.24	32.20	2.98
$R_{2}\_z$	32.80		1.86
$R_{3}\_z$	33.65		4.50

**Table 6 polymers-14-00293-t006:** The detailed results of SRPF-LB.

Sample (*z* = 1~10)	Average Measurements (μm)	Prediction (μm)	Error (%)
$R_{1}\_z$	5.40	5.80	6.89
$R_{2}\_z$	6.35		9.48
$R_{3}\_z$	6.41		10.52

**Table 7 polymers-14-00293-t007:** Parameters for testing sensitivity.

Case	Default	Lower Value	Upper Value
1. Extrusion width (mm)	0.4	0.3/0.2	-
2. Extrusion temperature (°C)	200	190	210
3. Layer thickness (mm)	0.15	0.1	0.2

**Table 8 polymers-14-00293-t008:** The influencing degree of the three processing parameters.

Parameters	Minimum Value	Maximum Value	The Rates of Surface Roughness Growth (%)
Extrusion width (mm)	0.2	0.4	−13.17
Extrusion temperature (°C)	190	210	−35.15
Layer thickness (mm)	0.1	0.2	56.74

## Data Availability

Data presented in this study is available upon request from the corresponding authors.

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
