# Peer review of "Theoretical and Experimental Investigation on the 3D Surface Roughness of Material Extrusion Additive Manufacturing Products"

_polymers, 2022, doi:10.3390/polym14020293_

Round 1

Reviewer 1 Report

The authors have presented new work on the surface roughness prediction of 3D printed components.  The paper can be published with minor corrections. 

1) section 3.1: Which 3D printing method did you use for printing the samples? Add this in the main script. 

2) Show the real photographs of the samples you printed. 

3) section 3.2: This section needs major rework. What standard did you follow for roughness measurement? ISO 4287, 88? ISO 25178-1,2,3?

4) What was the measurement area? The roughness depends on this area you measured and the settings that you used.

The authors should refer to the following paper and even cite more MDPI papers that discuss the relevant issue.

Nagalingam, A.P.; Vohra, M.S.; Kapur, P.; Yeo, S.H. Effect of Cut-Off, Evaluation Length, and Measurement Area in Profile and Areal Surface Texture Characterization of As-Built Metal Additive Manufactured Components. Appl. Sci. 202111, 5089. https://doi.org/10.3390/app11115089

5) Which roughness parameter did you measure? Ra, Rz? Does your model predict both of these parameters?

6) Table 4: "prediction" spelling error

7) Please state clearly in both Abstract and Conclusion that why someone has to use this model as compared to post-printing measurement of roughness. You state that it will be useful for future research, pls state what it is clearly. 

Author Response

Response to Reviewer 1 Comments

Point 1: section 3.1: Which 3D printing method did you use for printing the samples? Add this in the main script.

 Response 1: Thanks for your suggestion. The 3D printing method used in this article is material extrusion (ME) additive manufacturing technology. To clarify this, the ME equipment has been added in Figure 8 (a) of the manuscript. The ME device is shown in the following figure.

Point 2: Show the real photographs of the samples you printed.

Response 2: Thanks for your suggestion. The real photo of the samples has been added in Figure 8 (b) of the manuscript. As shown in the figure below.

Point 3: Section 3.2: This section needs major rework. What standard did you follow for roughness measurement? ISO 4287, 88? ISO 25178-1,2,3?

Response 3: Thanks for your comments. The standard for measuring surface roughness used in this paper is ISO 4287: 1997. To clarify this, the author added the corresponding introduction in Section 3.2 (as marked in yellow).

Point 4: What was the measurement area? The roughness depends on this area you measured and the settings that you used.

The authors should refer to the following paper and even cite more MDPI papers that discuss the relevant issue.

Nagalingam, A.P.; Vohra, M.S.; Kapur, P.; Yeo, S.H. Effect of Cut-Off, Evaluation Length, and Measurement Area in Profile and Areal Surface Texture Characterization of As-Built Metal Additive Manufactured Components. Appl. Sci. 2021, 11, 5089. https://doi.org/10.3390/app11115089

Response 4: Thanks for your question and suggestion. The author has referred to the above paper and introduced it in Section 3.2. “Through microscope observation, the surface roughness test was carried out on different typical positions (where the filaments bonded normally and no defects occurred) of the sample’s surface”.

Point 5: Which roughness parameter did you measure? Ra, Rz? Does your model predict both of these parameters?

Response 5: Thanks for your question. In this paper, the roughness parameter studied is Ra. The proposed model can predict the surface profile and Ra value of surface roughness.

Point 6: Table 4: "prediction" spelling error.

Response 6: Sorry for the carelessness. It has been corrected.

Point 7: Please state clearly in both Abstract and Conclusion that why someone has to use this model as compared to post-printing measurement of roughness. You state that it will be useful for future research, plsease state what it is clearly.

Response 7: Thanks for your comments. The author has added the explanation of the reason for establishing this model and its uses in both Introduction (2nd paragraph, marked in yellow) and Conclusion (the last sentence in the end), as well as the clear statement of its use in the future. The details are as follows: “The proposed model can help mitigate the time-consuming and expensive iterative experiments performed, and it can also provide the clue to find the effective method to improve the surface quality of ME components”.

The above is the author's response to the reviewers’ comments, please have a check.

Best Wishes,

Shijie Jiang

Reviewer 2 Report

Dear Authors,

By considering the mentioned comments, your paper would be an interesting paper based in the subjcet you are studying.

Best wishes

Author Response

Response to Reviewer 2 Comments

Point 1: The paper is well-organized and each section has been well defined, however, I propose to consider a double-check on the grammar.

Response 1: Thanks for your comments. The author has double-checked and revised the whole article.

Point 2: As you considered the neck-growth, there are several parameters that play a crucial role in this phenomenon. I propose you to consider adding a phrase explaining the effect of temperature and temperature dependence viscosity on the neck-growth during deposition of filaments. Temperature plays an important role and I have not seen any explanation in your literature review. In order to avoid delay in your research career, you can use the following references that have been performed by Vanaei et al. In these references, there is consequence of the effect of temperature on the bonding between filament as well as the effect of some process parameters on the temperature evolution of filament and their fatigue lifetime. Also you can find a review paper that explain the importance of in-process monitoring of temperature profile of filaments during 3D-printing as it is not well considered by researchers:

https://doi.org/10.1007/s10853-020-05057-9

https://doi.org/10.1002/pen.25555

https://doi.org/10.1108/RPJ-11-2019-0300

https://doi.org/10.3390/thermo1030021

The objective is not to just ask for citing the papers, but to enhance the level of your literature review to respect the subject you are studying. So, by taking into account the mentioned papers, you will have a better explanation of why you have performed this study

Response 2: Thanks for your suggestion and comment. The author has referred to the above references and introduced them in Section 1 (the yellow-marked contents in the 3rd and 4th paragraph). The details are as follows: “Due to the layer-by-layer manufacturing process, the forming quality of ME products was controlled by the thermal energy (temperature distribution) of the extruded material filaments [28, 29]. Vanaei et al [26, 27] studied the influence of extrusion temperature and the temperature between adjacent extruded filaments on the shape, size and fatigue life of ME products, separately. The experimental results showed that optimizing the above temperature was necessary for achieving the optimization of ME products’ forming quality. But little information is available in the literature mentioned above.”

Point 3: Based on the previous comment, I propose you to consider explaining that the temperature dependence viscosity is an important issue that exist in this process (equation 6).

Response 3: Thanks for your suggestion. The author has added the introduction that the temperature dependence viscosity is an important issue that exists in ME process in the paragraph below Equation (6). The details are as follows: “which is an important factor dependent on the extrusion temperature and the temperature between adjacent extruded filaments [26, 27 and 33].”

Point 4: Another remark goes to the geometry you have used. Have you think of existing different mechanisms (mostly heat transfer) during deposition of filaments?

Response 4: Thanks a lot for your question. Due to the layer-by-layer manufacturing process of ME technique, this paper mainly investigated the adhesion (forming process of bonding neck) between adjacent extruded filaments. It stops growing when the temperature of the extrudate drops to the critical temperature. Although the heat transfer problem is an important factor influencing the forming quality of ME products, it was not taken into account in this paper. But, the author will focus on it and the corresponding mechanism in the near future, not only due to its importance, but also its complexity.

The above is the author's response to the reviewers’ comments, please have a check.

Best Wishes,

Shijie Jiang

Round 2

Reviewer 2 Report

Best wishes in your work.

Author Response

Thanks for your comments. The author has checked the full text and corrected the spelling mistakes in the full text in the manuscript.